# Differences in cancer patients' work-cessation risk, based on gender and type of job: Examination of middle-aged and older adults in super-aged Japan

**Shuhei Kaneko**[1]*, **Haruko Noguchi**[2], **Rong Fu**[2], **Cheolmin Kang**[2], **Akira Kawamura**[2], **Shinsuke Amano**[3], **Atsushi Miyawaki**[4]

1 Graduate School of Economics, Waseda University, Tokyo Japan, 2 Faculty of Political Science and Economics, Waseda University, Tokyo Japan, 3 Japan Federation of Cancer Patient Groups, Yokohama, Japan, 4 Graduate School of Medicine, the University of Tokyo, Japan

* shuhei7700@ruri.waseda.jp

## Abstract

### Objectives

In this paper, we aim to estimate the effect cancer diagnosis has on labour-force participation among middle-aged and older populations in Japan. We investigate the impact of cancer diagnosis on job cessation and the gap between gender or job types.

### Methods

We sourced data from a nationwide, annual survey targeted population aged 51–70 featuring the same cohort throughout, and examined respondents' cancer diagnoses and whether they continued to work, while also considering differences between gender (observations: 53 373 for men and 44 027 for women) and occupation type (observations: 64 501 for cognitive worker and 20 921 for manual worker) in this regard. We also examined one-year lag effects, using propensity score matching to control for confounding characteristics. We also implement Logistic regression and derive the odds ratio to evaluate the relative risk of cancer diagnosis, which supplements the main result by propensity score matching.

### Results

Overall, the diagnosis of cancer has a huge effect on labour-force participation among the population, but this effect varies across subpopulations. Male workers are more likely to quit their job in the year they are diagnosed with cancer (10.1 percentage points), and also in the following year (5.0 percentage points). Contrastingly, female workers are more likely to quit their job immediately after being diagnosed with cancer (18.6 percentage points); however, this effect totally disappears when considering likelihoods for the following year. Cognitive workers are more prone to quit their job in the year of diagnosis by 11.6 percentage points, and this effect remains significant, 3.8 percentage points, in the following year. On the other

**Data Availability Statement:** Data cannot be shared publicly because of Article 33 of Japan's Statistics Act (Act No. 53 of 2007). Data are

available based on the criteria set by the Ministry of Health, Labour, and Welfare (contact via https://www.mhlw.go.jp/toukei/sonota/chousahyo.html) for researchers.

**Funding:** HN is funded by funded by the Japanese Ministry of Health, Labour and Welfare (H29-Junkankitou-Ippan-002), and Organization for University Research Initiatives of Waseda University. RF is funded by Japan Society for the Promotion of Science (JSPS KAKENHI Grant Number 17H07182). The funders had no role in study design, data collection and analysis, decision to publish, or preparation of the manuscript.

**Competing interests:** The authors have declared that no competing interests exist.

hand, for manual workers the effect during the year of diagnosis is huge. It amounts to 18.7 percentage points; however, the effect almost disappears in the following year.

## Conclusion

Our results indicate the huge effect of cancer on job cessation, and that there might be a degree of discrimination in workplaces between gender and job types.

## Introduction

Cancer is one of the most prevalent diseases worldwide. Approximately 14 million people are newly diagnosed each year and, in 2012, eight million people died from cancer-related causes [1]. In addition to its severe impact on individuals, cancer imposes a substantial economic burden on societies. The prevention and treatment of cancer are extremely expensive; furthermore, some patients are unable to continue working, and are, consequently, forced to depend on financial, social, and mental supports from other family members or friends [2]. Several studies have shown that continued participation in the labour market can benefit cancer patients, not only from a financial aspect, but also as communication with colleagues or friends can fostering personal satisfaction and/or healthy distraction [3–6]. For this reason, governments in developed countries and regions have recently begun efforts to construct a system that supports cancer patients in regard to securing jobs and protecting them from discrimination in their workplaces [7].

In Japan, as well as other developed countries, cancer has been and is one of the most critical issues in society and workplaces. Cancer accounts for almost 30% of total number of death in 2017 [8] and it has been the leading cause of death in Japan since 1981 [9]. Also, data in 2012 indicates that a third of total cancer patients were diagnosed as cancer when they were younger than 65 years old [10]. We should note that 325 thousand cancer patients regularly go to hospital for cancer treatment while they are working [10].

Given this situation, the Ministry of Health, Labour, and Welfare (MHLW) has established guidelines for helping cancer patients receive necessary medical treatments while continuing to work [11], which is based on the Basic Plan to Promote Cancer Control Programs revised in 2012. This plan aims to enhance the cancer control among working population and children. Considering these developments, it is clear that exploring the relationship between the onset of cancer and working status has recently become a major concern for policymakers.

Numerous studies have confirmed the various negative impacts of cancer on working status; for example, a cancer diagnosis is associated with lower income [12,13], lower labour participation [14,15], and workplace discrimination [16,17]. However, some of these studies may have suffered from low data representativeness, while others merely observed the correlation between cancer and employment status, without considering the causal inferences. There are many confounding factors related to the onset of cancer, and identifying a means of overcoming these analytical difficulties and obtaining statistically unbiased results from which we develop both scientific and appropriate policy implications is an extremely daunting challenge. To this end, in the present study, in which we sought to determine whether the risk of work cessation after cancer diagnosis was impacted by gender and job type, we applied a frequently used econometric strategy, the propensity score matching (PSM) method, in our examination of data from a nationwide population-based longitudinal survey conducted in Japan. PSM is a commonly used statistical strategy to assign individuals seemingly at random into 'treatment

group' and 'control group', by one-by-one matching (or one-to-many matching) individuals with similar risk based on various observed characteristics. In addition to this, to evaluate the relative risk of job cessation between cancer patients and non-cancer patients, we implement multivariate logistic regression and derive the odds ratio.

This study may contribute to the field in three ways. First, to the best of our knowledge, this is the first study to quantitatively examine whether the risk of work cessation after a cancer diagnosis is impacted by gender and job type. Second, we focus on middle-aged and older persons aged 50–71 years in a super-aged society: Japan. The risk of cancer incidence begins to increase from middle-aged strata, and its risk goes up as the passage of age [18]. In fact, the incidence risk of cancer by 69 years old is 20.1% for male and 17.6% for female in Japan. Considering that the same statistic is only 2.4% (male) and 5.2% (female) by 49 years old, the population we focused should be mooted regarding cancer. A similar problem occurs in other developed countries and regions [19,20], where declines in the labour force, as well as reductions in financial resources for healthcare as a result of population aging and decreasing birth rates, is becoming a serious problem. Finally, our study provides reliable scientific evidence regarding an Asian country, Japan, which is novel because most previous related studies have been conducted in Western countries such as the United States, Australia, and Northern European countries. In Japan, a survey focusing on the cancer diagnosis and occupation was conducted in 2004, where 34% of people who were newly diagnosed as cancer reported that they quit their job or they were dismissed [10]. However, now that this survey becomes outdated, it is obvious that re-evaluating the impact is critical and the rigorous statistical analyses which we provide will help a deeper understanding of this issue. Specifically, the importance of our study can be summarized as follows: not merely we clarify the relationship between cancer diagnosis and job cessation, but also we focus on the heterogeneity of the effects based on gender and types of job, applying the econometric strategy to identify the causal effect.

The research purpose of the present study is to investigate "how large is the effect of cancer diagnosis on job cessation?" and "is there any gap between gender or job types?" To the best of our knowledge, we are first to tackle this question and the resulting biases in Japanese society. Considering that gender or work style-oriented discrimination at the workplace is still prevailing worldwide, our results should contribute understanding these discriminations and improving them.

## Methods

### Data

Data for this study were sourced from a nationwide population-based longitudinal survey, the 'Longitudinal Survey of Middle-aged and Elderly Persons' (LSMEP), which has been conducted annually by MHLW since 2005. For this research, we used all data from 2005 up to the latest available year, 2016. A previous administrative survey ('the Comprehensive Survey of Living Conditions') was conducted in 2004, and examined 5,280 districts in Japan; the LSMEP randomly selected, through a two-stage sampling procedure, 2,515 of these districts. From these districts, 40,877 individuals aged 50–59 years at the end of October 2005 were selected, with the number chosen from each district being in proportion to the entire population of the district (with regard to age and sex distribution). Of these individuals, 33,185 successfully responded (response rate: 82.7%), and these represented the baseline sample, and were followed up thereafter. In the subsequent surveys, questionnaires were delivered to the individuals who had responded within the previous two years. Questionnaire sheets were initially delivered to each household by enumerators; however, since the sixth survey in 2010, the

questionnaires have been sent by postal mail. No respondents have been newly recruited; therefore, the response rate had decreased to 53.6% (21,916 respondents) by 2016.

We obtained official permission to use LSMEP from the MHLW (Tohatsu-0507-3 on May 7, 2018) on the basis of Article 32 of the Statistics Act. Ethical reviews of these data were not required, in accordance with the 'Ethical Guidelines for Medical and Health Research Involving Human Subjects' of the Japanese government [21].

### Study settings

To capture the causal effect of the 'health shock' associated with the diagnosis of cancer and its impact on the risk of work cessation, we applied PSM in two study settings: one-year lagged and simultaneous, similar strategy was implemented by García-Gómez (2011) [22], which are described in Fig 1. For the one-year lagged setting shown in Panel (A), we applied the following four steps: (i) We defined a sequence of three-year time-windows for the entire survey period (2005–2016) (e.g., $t = 1$, $t = 2$, and $t = 3$), which yielded 10 time-windows (2005–2007, 2006–2008, 2007–2009, . . ., 2014–2016). (ii) For each of the 10 time-windows, we extracted respondents who were working in the labour market at both $t = 1$ and $t = 2$, and also those who, at $t = 1$, had never previously been diagnosed with cancer. (iii) For each time-window,

Panel (A): Procedure (1) to (4) [one-year lagged]

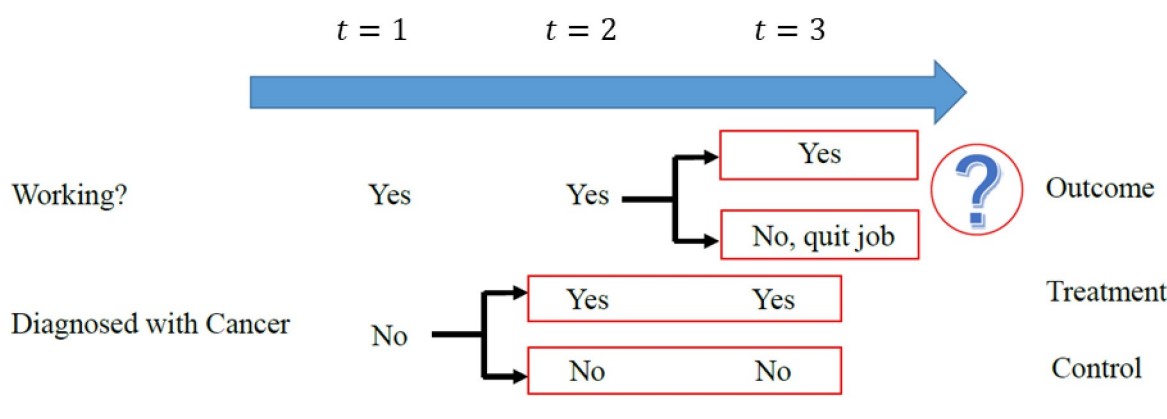

Panel (B): Procedure (5) to (8) [simultaneous]

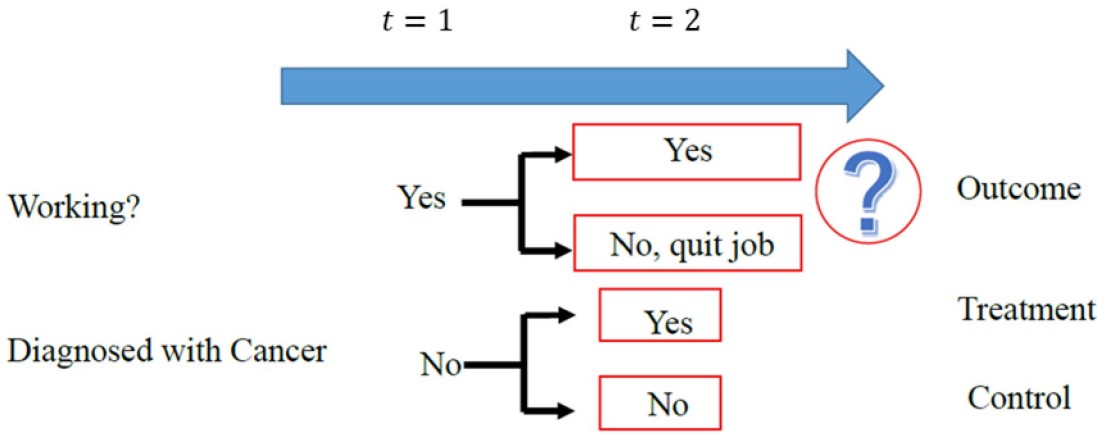

**Fig 1. Two types of study settings for PSM.** For both cases, PSM was executed with regard to the characteristics at $t = 1$.

we allocated those who were diagnosed with cancer at $t = 2$ to the 'treatment group', and those who had not been diagnosed with cancer by the end of each time-window to the 'control group'. (iv) The outcome measure was a dichotomous variable, which took '1' if a respondent quit his/her job at $t = 3$, and '0' otherwise.

In the first strategy, we were not concerned about 'reverse causality' between cancer diagnosis and job-quitting behaviour, because we restricted our sample to those who continued to work until $t = 2$, even after they had been diagnosed with cancer. In other words, unless the data were misreported, these people could not have quit their jobs before being diagnosed with cancer. However, since individuals who both quit their jobs and were diagnosed with cancer at $t = 2$ were excluded from the estimate, which might cause 'selection bias,' the true effect could have been underestimated. Therefore, we also applied the simultaneous setting, shown in (v)–(viii) in Panel (B): (v) We defined a sequence of two-year time-windows for the entire survey period ($t = 1$ and $t = 2$), which yields 11 time-windows (2005–2006, 2006–2007, 2007–2008, . . ., 2015–2016); (vi) For each of the 11 time-windows, we chose respondents who were working in the labour market at $t = 1$ and also those who had never previously been diagnosed with cancer at $t = 1$. (vii) This step was identical to (iii), above. (viii) The outcome measure was a dichotomous variable, which took '1' if a respondent quit his/her job at $t = 2$, and '0' otherwise.

In this second strategy, the estimates could capture the simultaneous effect of a cancer diagnosis while avoiding 'selection bias'. However, the 'reverse causality' between the onset of cancer and job-quitting behaviour could not be completely avoided, as these seem to be determined simultaneously at $t = 2$. Given the trade-off between the two study settings, we show both results in Fig 1 (Panel (A) and Panel (B)).

The procedure of sample selection seems to be complicated; and therefore, we provide the sample selection flow chart by Fig 2 to help readers to grasp what is going on in this study.

## Propensity score matching

We applied the PSM method to balance, between the treatment and control groups, various confounding factors at the baseline ($t = 1$) of each time-window that should not be affected by the diagnosis of cancer at $t = 2$. Further, a Probit model was used to evaluate the propensity scores of the risk of being diagnosed with cancer (treatment), considering various individual characteristics: age, marital status, educational achievement, self-rated health status (SRH), psychological distress (measured using Kessler 6; K6), number of children living in the same household, household size, ability of daily living (ADL), degree of daily exercise, alcohol and smoking behaviour, logarithm of individual income in the last month, logarithm of the sum of the individual's and his/her spouse's income in the last month, vocational category, diagnosis of diseases other than cancer (such as hypertension and dyslipidaemia), and residential location. Note that we exclude self-employed individuals and family workers throughout our analysis; this was in order to identify 'exit from the labour market' as clearly as possible. Table 1 shows the definition of all the variables considered in the PSM procedure.

PSM was estimated separately in terms of gender (**Model 1**: male versus female) and type of job (**Model 2**: cognitive versus manual). Regarding type of job, we classified workers as 'cognitive workers' if they engaged in administrative or managerial, professional, clerical, sales, or service work; and as 'manual workers' if they engaged in security, agriculture, forestry, fishery, manufacturing process, transport and machine operation, construction, or transportation work. Then, finally, we estimated the average treatment effect on the treatment group (ATT). ATT is calculated as the difference in the probability of job cessation between people

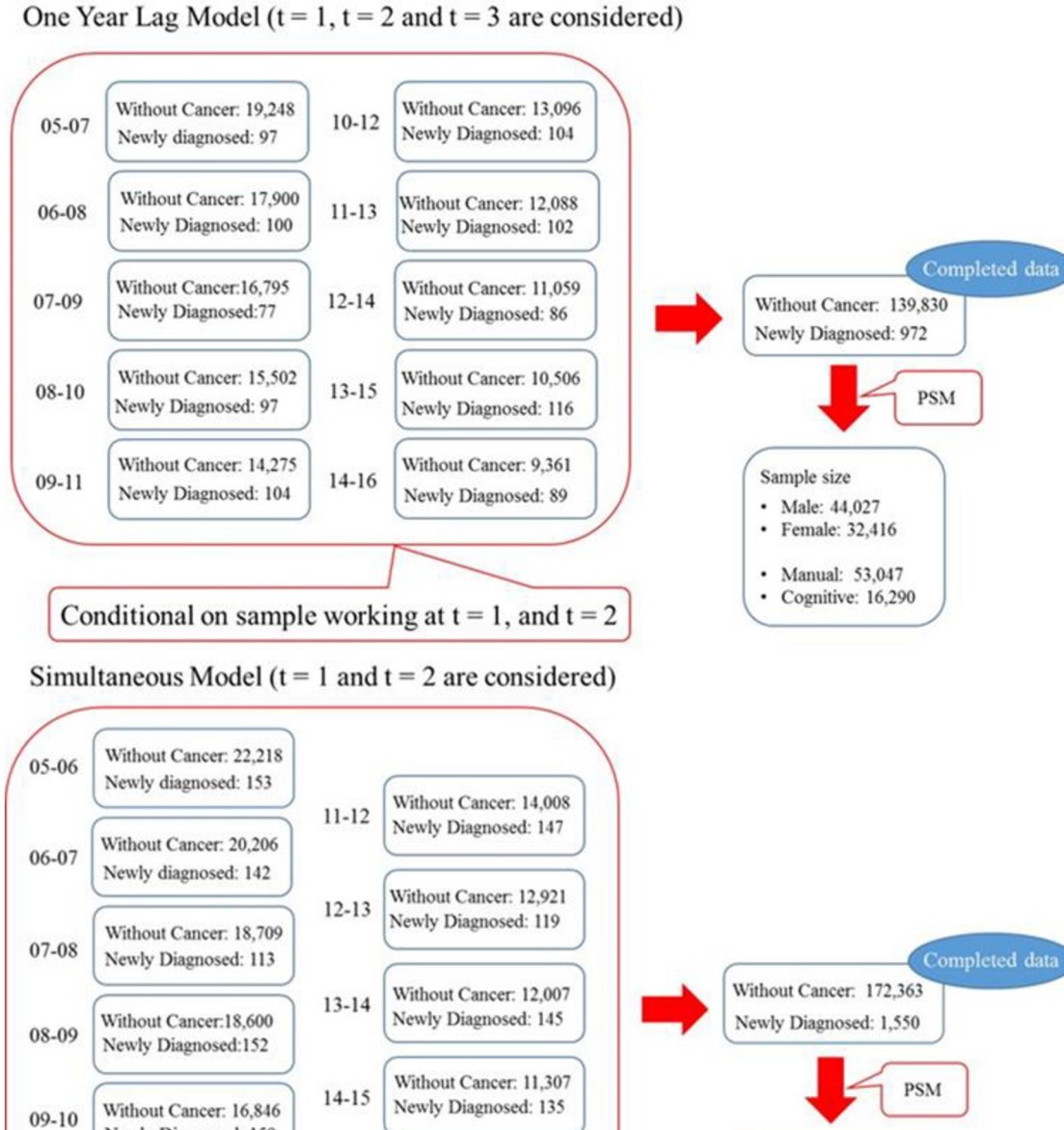

**Fig 2. Sample selection flow chart.**

diagnosed as cancer and non-diagnosed population matched by PSM based on the covariates. If we denote them as $P_{Diag}$ and $P_{NonDiag}$, then

$$ATT = P_{Diag} - P_{NonDiag}. \tag{1}$$

**Table 1. Definition of variables.**

| Variable | Definition |
|---|---|
| *Age* | Individual's age |
| *Marital status* | |
| Married | = 1 if he/she is married; otherwise = 0 |
| Divorced/widowed | = 1 if he/she is divorced or widowed; otherwise = 1 |
| Single | = 1 if he/she is single; otherwise = 2 |
| *Education* | |
| Higher than univ. level | = 1 if he/she has university bachelor or higher degree; otherwise = 0. |
| *SRH* | Self-rated health status is classified into 6 categories; excellent, good, comparatively good, comparatively bad, bad, and very bad. |
| *K6 score* | The score is measured through six items: "During the past 1 month, did you feel (i) nervous, (ii) hopeless, (iii) restless or fidgety, (iv) so depressed that nothing could cheer you up, (v) that everything was an effort, and (vi) worthless?" For each question, the response option varies from "none of the time" (yielding a score of 0) to "all of the time" (yielding a score of 4), which means the total score has a range of 0–24. |
| *Household character* | |
| # of children in HH | a total number of children resided with individual. |
| HH size | total number of household members |
| Ability of daily activity | = 1 if he/she answered that they suffer doing daily activity due to the health issue; otherwise = 0. |
| *Exercise* | |
| Mild | Frequency of mild exercise (e.g. stretching) is indexed into six categories; 0 = not at all, 1 = once a month, 2 = once a week, 3 = two or three times a week, 4 = four or five times a week, and 5 = almost every day. |
| Moderate | Frequency of moderate exercise (e.g. walking, jogging) is indexed into six categories; 0 = not at all, 1 = once a month, 2 = once a week, 3 = two or three times a week, 4 = four or five times a week, and 5 = almost every day. |
| Hard | Frequency of hard exercise (e.g. aerobics, swimming) is indexed into six categories; 0 = not at all, 1 = once a month, 2 = once a week, 3 = two or three times a week, 4 = four or five times a week, and 5 = almost every day. |
| *Alcohol* | The average amount of alcohol consumed (calculated in terms of Japanese sake) when drinking was determined using the following categories: 1 = less than one cup/glass (180 ml), 2 = one to three glass(es), 3 = three to five glasses, 4 = five glasses or more. 0 was allocated to those who did not usually drink (or could not). |
| *Smoking* | Average number of cigarettes smoked per day was indexed as follows; 0 = none at all, 1 = 10 or less, 2 = 11 to 20, 3 = 21 to 30, 4 = more than 30. |
| *Log of income* | The logarithm of an individual's monthly income (unit: JPY 10,000) in the last year. |
| *Log of HH income* | The logarithm of sum of household members' monthly income (unit: JPY 10,000) in the last year. |
| *Job category* | |
| Professional | = 1 if an individual engages in a professional job; otherwise 0. |
| Managerial | = 1 if an individual engages in a managerial job; otherwise 0. |
| Clerical | = 1 if an individual serves as a clerical worker; otherwise 0. |
| Sales | = 1 if an individual engages in a sales job; otherwise 0. |
| Service | = 1 if an individual serves as a service worker; otherwise 0. |
| Security | = 1 if an individual serves as a security worker; otherwise 0. |
| Primary industries | = 1 if an individual engages in primal industries (including agriculture, fishery industry, and forest industry); otherwise 0. |
| Transport | = 1 if an individual engages in a transportation job; otherwise 0. |
| Manufacturing | = 1 if an individual engages in a manufacturing job; otherwise 0. |
| *Risk Factor* | |
| Hypertension | = 1 if an individual was diagnosed with hypertension when t = 1; otherwise 0. |
| Dyslipidaemia | = 1 if an individual was diagnosed with dyslipidaemia when t = 1; otherwise 0. |

Throughout the paper, nearest neighbour matching was applied, through which each cancer patient was matched with a weighted average of the 15 closest non-cancer patients in terms of the propensity score, and we imposed a calliper of 0.01 of the score to avoid poor matching balances. Therefore, $P_{NonDiag}$ is a weighted probability of job cessation of non-diagnosed and matched population, that is, $P_{NonDiag} = \sum_{i=1}^{15} w_i QUIT_i$, where $w_i$ is weight for each matched sample and $QUIT_i$ is a dichotomous variable indicating job cessations. Because ATT is the simple difference in two probabilities, ATT is measured by the changes in *percentage points*.

Further rigorous discussion of PSM methodology, such as our matching quality, is shown in the web appendix [23,24]. All analyses (including Logistic regression) were performed using Stata MP 15.1. Almost all of the PSM results were performed by a user-written program, psmatch2.

### Logistic regression

In the logistic regression, the probability of job cessation ($p(quit)$)) is formalized as

$$p(quit) = \frac{\exp(\beta_0 + \beta_1 Diag_{it} + \gamma X_{it} + \delta\theta_t + \epsilon_{it})}{1 + \exp(\beta_0 + \beta_1 Diag_{it} + \gamma X_{it} + \delta\theta_t + \epsilon_{it})} \ , \tag{2}$$

where $Diag_{it}$ is a dichotomous variable indicating the diagnosis of cancer, $X_{it}$ is a set of other controlled variables, $\theta_t$ is the year fixed effect, and $\epsilon_{it}$ denotes the random error term.

Considering that the odds is defined as $p(quit) / 1 - p(quit)$, the odds ratio between diagnosed and non-diagnosed population can be expressed as

$$OR = \frac{p(quit)/1 - p(quit)| \, Diag = 1}{p(quit)/1 - p(quit)| \, Diag = 0} = \frac{\exp\left(\log\left(p(quit)/1 - p(quit)| \, Diag = 1\right)\right)}{\exp\left(\log\left(p(quit)/1 - p(quit)| \, Diag = 1\right)\right)} \tag{3}$$

$$= \frac{\exp(\beta_0 + \beta_1 * 1 + \gamma X_{it} + \delta\theta_t + \epsilon_{it})}{\exp(\beta_0 + \beta_1 * 0 + \gamma X_{it} + \delta\theta_t + \epsilon_{it})} = \exp(\beta_1).$$

In each regression, all the explanatory variables listed in the descriptive tables (explained in the next section) are included as $X_{it}$ to control for the effects caused by observable predictors.

### Ethics statement

The data of LSMEP is publicly available for any researchers, as long as they obtain official permission from the Ministry of Health, Labour and Welfare (MHLW) through the application procedure on the Statistics Act (Article 32 & 33) in Japan. All data were fully anonymized before we access them and the ethics committee at Waseda University waived the requirement for informed consent. (approval no. 729–420).

## Results

### Descriptive results of simultaneous setting

After the procedure described in Section 3.1, we reached 173,913 (Diagnosed: 1,550, Not diagnosed: 172,363) number of sample in simultaneous setting. Among them, 53,373 male workers, 44,027 female workers, 64,501 cognitive workers, and 20,921 manual workers are finally exploited for PSM estimation. Tables 2 and 3 show the result of mean comparisons and statistical tests between the treatment (those who were diagnosed with cancer) and control (those who were not diagnosed with cancer) groups. The result before PSM showed that the means of

**Table 2. Descriptive statistics before and after PSM for Model 1.** (simultaneous setting).

| | | Male | | | | Female | | | |
| --- | --- | --- | --- | --- | --- | --- | --- | --- | --- |
| | | Mean | | t-test/$\chi^2$ test | | Mean | | t-test/$\chi^2$ test | |
| | | Diagnosed | Not diag. | statistics | p-value | Diagnosed | Not diag. | statistics | p-value |
| Female | U | | | | | | | | |
| | M | | | | | | | | |
| Age | U | 60.163 | 58.592 | 8.79 | 0.000 | 58.547 | 58.332 | 0.95 | 0.344 |
| | M | 60.163 | 60.154 | 0.04 | 0.968 | 58.547 | 58.537 | 0.03 | 0.974 |
| Marital status | | | | | | | | | |
| Married | U | 0.921 | 0.899 | 2.69[+] | 0.101 | 0.802 | 0.792 | 0.18[+] | 0.669 |
| | M | 0.921 | 0.926 | −0.29 | 0.774 | 0.802 | 0.796 | 0.20 | 0.844 |
| Divorced/widowed | U | 0.052 | 0.054 | 0.04[+] | 0.851 | 0.145 | 0.169 | 1.34[+] | 0.248 |
| | M | 0.052 | 0.050 | 0.13 | 0.895 | 0.145 | 0.146 | −0.05 | 0.958 |
| Single | U | 0.027 | 0.047 | 4.57[+] | 0.033 | 0.053 | 0.039 | 1.80[+] | 0.179 |
| | M | 0.027 | 0.024 | 0.30 | 0.763 | 0.053 | 0.058 | −0.26 | 0.792 |
| Educational achievement | | | | | | | | | |
| Higher than univ. level | U | 0.279 | 0.323 | 4.53[+] | 0.033 | 0.085 | 0.078 | 0.19[+] | 0.662 |
| | M | 0.279 | 0.273 | 0.20 | 0.843 | 0.085 | 0.088 | −0.14 | 0.888 |
| Self-rated health status (SRH) | | | | | | | | | |
| Excellent | U | 0.046 | 0.058 | 1.42[+] | 0.233 | 0.041 | 0.058 | 1.68[+] | 0.195 |
| | M | 0.046 | 0.046 | −0.02 | 0.984 | 0.041 | 0.042 | −0.08 | 0.937 |
| Good | U | 0.262 | 0.340 | 14.14[+] | 0.000 | 0.239 | 0.342 | 14.80[+] | 0.000 |
| | M | 0.262 | 0.259 | 0.08 | 0.940 | 0.239 | 0.246 | −0.21 | 0.834 |
| Comparatively good | U | 0.435 | 0.446 | 0.26[+] | 0.608 | 0.509 | 0.469 | 2.02[+] | 0.155 |
| | M | 0.435 | 0.441 | −0.20 | 0.838 | 0.509 | 0.500 | 0.23 | 0.820 |
| Comparatively bad | U | 0.223 | 0.131 | 37.63[+] | 0.000 | 0.173 | 0.114 | 10.85[+] | 0.001 |
| | M | 0.223 | 0.220 | 0.10 | 0.917 | 0.173 | 0.177 | −0.12 | 0.906 |
| Bad | U | 0.029 | 0.021 | 1.62[+] | 0.203 | 0.035 | 0.015 | 7.83[+] | 0.005 |
| | M | 0.029 | 0.028 | 0.11 | 0.911 | 0.035 | 0.033 | 0.09 | 0.930 |
| Very bad | U | 0.006 | 0.003 | 0.89[+] | 0.346 | 0.003 | 0.002 | 0.31[+] | 0.579 |
| | M | 0.006 | 0.005 | 0.14 | 0.888 | 0.003 | 0.001 | 0.44 | 0.660 |
| K6 score | U | 2.821 | 2.677 | 0.91 | 0.362 | 3.651 | 3.129 | 2.44 | 0.015 |
| | M | 2.821 | 2.786 | 0.15 | 0.878 | 3.651 | 3.670 | −0.06 | 0.953 |
| # of children in household | U | 0.704 | 0.828 | −3.15 | 0.002 | 0.623 | 0.734 | −2.42 | 0.016 |
| | M | 0.704 | 0.702 | 0.04 | 0.969 | 0.623 | 0.616 | 0.10 | 0.917 |
| Household size | U | 3.058 | 3.167 | −1.82 | 0.069 | 2.934 | 2.990 | −0.71 | 0.481 |
| | M | 3.058 | 3.071 | −0.16 | 0.871 | 2.934 | 2.921 | 0.11 | 0.909 |
| Ability of daily activity,[(i)] | U | 0.083 | 0.051 | 10.67[+] | 0.001 | 0.107 | 0.082 | 2.70[+] | 0.101 |
| | M | 0.083 | 0.079 | 0.23 | 0.820 | 0.107 | 0.104 | 0.13 | 0.897 |
| Exercise[(ii)] | | | | | | | | | |
| Mild | U | 1.398 | 1.366 | 0.38 | 0.703 | 1.516 | 1.487 | 0.27 | 0.785 |
| | M | 1.398 | 1.395 | 0.02 | 0.980 | 1.516 | 1.526 | −0.07 | 0.943 |
| Moderate | U | 1.083 | 1.076 | 0.10 | 0.922 | 0.956 | 0.949 | 0.08 | 0.933 |
| | M | 1.083 | 1.124 | −0.40 | 0.688 | 0.956 | 0.956 | 0.00 | 0.999 |
| Hard | U | 0.181 | 0.179 | 0.06 | 0.953 | 0.204 | 0.195 | 0.23 | 0.818 |
| | M | 0.181 | 0.169 | 0.28 | 0.783 | 0.204 | 0.213 | −0.14 | 0.887 |
| Alcohol[(iii)] | U | 1.464 | 1.367 | 2.29 | 0.022 | 0.472 | 0.439 | 0.83 | 0.406 |
| | M | 1.464 | 1.467 | −0.06 | 0.949 | 0.472 | 0.462 | 0.18 | 0.859 |

(*Continued*)

**Table 2.** (Continued)

| | | Male | | | | Female | | | |
|---|---|---|---|---|---|---|---|---|---|
| | | Mean | | t-test/χ² test | | Mean | | t-test/χ² test | |
| | | Diagnosed | Not diag. | statistics | p-value | Diagnosed | Not diag. | statistics | p-value |
| Smoking[iv] | U | 0.973 | 0.867 | 1.93 | 0.054 | 0.160 | 0.185 | −0.75 | 0.453 |
| | M | 0.973 | 0.968 | 0.06 | 0.954 | 0.160 | 0.145 | 0.36 | 0.718 |
| Log (income) | U | 3.319 | 3.424 | −3.28 | 0.001 | 2.552 | 2.507 | 0.99 | 0.324 |
| | M | 3.319 | 3.321 | −0.03 | 0.977 | 2.552 | 2.563 | −0.17 | 0.868 |
| Log (Household income) | U | 3.846 | 3.838 | 0.32 | 0.748 | 3.725 | 3.720 | 0.15 | 0.884 |
| | M | 3.846 | 3.853 | −0.19 | 0.847 | 3.725 | 3.742 | −0.28 | 0.777 |
| Job category | | | | | | | | | |
| Professional | U | 0.225 | 0.240 | 0.65[+] | 0.421 | 0.176 | 0.173 | 0.03[+] | 0.872 |
| | M | 0.225 | 0.218 | 0.27 | 0.784 | 0.176 | 0.173 | 0.10 | 0.922 |
| Managerial | U | 0.198 | 0.189 | 0.27[+] | 0.601 | 0.028 | 0.028 | 0.00[+] | 0.997 |
| | M | 0.198 | 0.195 | 0.13 | 0.897 | 0.028 | 0.030 | −0.13 | 0.900 |
| Clerical | U | 0.098 | 0.098 | 0.00[+] | 0.994 | 0.173 | 0.189 | 0.50[+] | 0.479 |
| | M | 0.098 | 0.098 | 0.02 | 0.983 | 0.173 | 0.179 | −0.21 | 0.835 |
| Sales | U | 0.056 | 0.061 | 0.23[+] | 0.632 | 0.138 | 0.103 | 4.38[+] | 0.036 |
| | M | 0.056 | 0.056 | 0.01 | 0.993 | 0.138 | 0.132 | 0.25 | 0.805 |
| Service | U | 0.085 | 0.078 | 0.33[+] | 0.568 | 0.217 | 0.222 | 0.06[+] | 0.814 |
| | M | 0.085 | 0.090 | −0.29 | 0.775 | 0.217 | 0.209 | 0.25 | 0.802 |
| Security | U | 0.033 | 0.037 | 0.28[+] | 0.595 | 0.003 | 0.001 | 0.95[+] | 0.329 |
| | M | 0.033 | 0.036 | −0.29 | 0.768 | 0.003 | 0.004 | −0.18 | 0.859 |
| Primary industries[v] | U | 0.017 | 0.012 | 1.07[+] | 0.301 | 0.016 | 0.011 | 0.83[+] | 0.363 |
| | M | 0.017 | 0.019 | −0.19 | 0.852 | 0.016 | 0.015 | 0.11 | 0.914 |
| Transport | U | 0.081 | 0.078 | 0.07[+] | 0.794 | 0.009 | 0.006 | 0.70[+] | 0.402 |
| | M | 0.081 | 0.080 | 0.03 | 0.976 | 0.009 | 0.009 | 0.00 | 1.000 |
| Manufacturing | U | 0.133 | 0.144 | 0.54[+] | 0.461 | 0.120 | 0.142 | 1.32[+] | 0.250 |
| | M | 0.133 | 0.134 | −0.05 | 0.961 | 0.120 | 0.125 | −0.22 | 0.828 |
| Risk factor | | | | | | | | | |
| Hypertension | U | 0.431 | 0.351 | 14.54[+] | 0.000 | 0.346 | 0.259 | 12.35[+] | 0.000 |
| | M | 0.431 | 0.442 | -0.37 | 0.708 | 0.346 | 0.344 | 0.04 | 0.965 |
| Dyslipidaemia | U | 0.317 | 0.248 | 13.25[+] | 0.000 | 0.299 | 0.234 | 7.44[+] | 0.007 |
| | M | 0.317 | 0.319 | -0.04 | 0.965 | 0.299 | 0.296 | 0.08 | 0.940 |

In the table, U denotes unmatched, and M denotes matched.

[+]: Chi-square statistics are reported. (Degree of freedom is 1)

[i] The ability of daily living was measured using a dichotomous variable that took the value of '1' if the respondent answered 'yes' to the question: 'do you have any problem in your daily life?'

[ii] Frequency of exercise is indexed into six categories; 0 = not at all, 1 = once a month, 2 = once a week, 3 = two or three times a week, 4 = four or five times a week, and 5 = almost every day.

[iii] The average amount of alcohol consumed (calculated in terms of Japanese sake) when drinking was determined using the following categories: 1 = less than one cup/ glass (180 ml), 2 = one to three glass(es), 3 = three to five glasses, 4 = five glasses or more. 0 was allocated to those who did not usually drink (or could not).

[iv] Average number of cigarettes smoked per day was indexed as follows; 0 = none at all, 1 = 10 or less, 2 = 11 to 20, 3 = 21 to 30, 4 = more than 30.

[v] Primary industries include agriculture, forestry, and fishery.

some characteristics systematically differed between the treatment and control groups in terms of some sub-samples, such as age, marital status, education level, SRH, K6 score, number of children, household size, ability of daily activity, engagement in health-related risk behaviours such as drinking alcohol and smoking, risks of being hypertension, and dyslipidaemia.

**Table 3. Descriptive statistics before and after PSM for Model 2.** (simultaneous setting).

| | | Cognitive | | | | Manual | | | |
|---|---|---|---|---|---|---|---|---|---|
| | | Mean | | t-test/$\chi^2$test | | Mean | | t-test/$\chi^2$test | |
| | | Diagnosed | Not diag. | statistics | p-value | Diagnosed | Not diag. | statistics | p-value |
| Female | U | 0.404 | 0.449 | 4.76[+] | 0.029 | 0.255 | 0.309 | 2.46[+] | 0.117 |
| | M | 0.404 | 0.403 | 0.04 | 0.968 | 0.255 | 0.255 | 0.02 | 0.987 |
| Age | U | 59.236 | 58.318 | 0.00 | 0.900 | 60.196 | 58.578 | 5.45 | 0.000 |
| | M | 59.236 | 59.195 | 0.17 | 0.862 | 60.196 | 60.227 | −0.07 | 0.942 |
| Marital status | | | | | | | | | |
| Married | U | 0.879 | 0.858 | 2.03[+] | 0.154 | 0.875 | 0.848 | 1.03[+] | 0.311 |
| | M | 0.879 | 0.880 | −0.05 | 0.957 | 0.875 | 0.882 | −0.19 | 0.849 |
| Divorced/widowed | U | 0.083 | 0.101 | 2.09[+] | 0.148 | 0.087 | 0.100 | 0.32[+] | 0.571 |
| | M | 0.083 | 0.083 | 0.01 | 0.994 | 0.087 | 0.086 | 0.05 | 0.961 |
| Single | U | 0.038 | 0.041 | 0.10[+] | 0.756 | 0.038 | 0.052 | 0.76[+] | 0.383 |
| | M | 0.038 | 0.037 | 0.08 | 0.934 | 0.038 | 0.033 | 0.26 | 0.793 |
| Educational achievement | | | | | | | | | |
| Higher than univ. level | U | 0.255 | 0.276 | 1.26[+] | 0.262 | 0.082 | 0.080 | 0.01[+] | 0.944 |
| | M | 0.255 | 0.258 | −0.13 | 0.893 | 0.082 | 0.082 | −0.01 | 0.990 |
| Self-rated health status (SRH) | | | | | | | | | |
| Excellent | U | 0.059 | 0.063 | 0.17[+] | 0.679 | 0.005 | 0.046 | 6.88[+] | 0.009 |
| | M | 0.059 | 0.060 | −0.08 | 0.934 | 0.005 | 0.008 | −0.26 | 0.796 |
| Good | U | 0.255 | 0.353 | 24.07[+] | 0.000 | 0.255 | 0.314 | 2.87[+] | 0.090 |
| | M | 0.255 | 0.255 | 0.00 | 0.996 | 0.255 | 0.274 | −0.39 | 0.695 |
| Comparatively good | U | 0.459 | 0.447 | 0.37[+] | 0.544 | 0.484 | 0.475 | 0.05[+] | 0.821 |
| | M | 0.459 | 0.454 | 0.19 | 0.853 | 0.484 | 0.479 | 0.08 | 0.934 |
| Comparatively bad | U | 0.194 | 0.118 | 31.41[+] | 0.000 | 0.217 | 0.139 | 9.25[+] | 0.002 |
| | M | 0.194 | 0.200 | −0.26 | 0.798 | 0.217 | 0.207 | 0.25 | 0.806 |
| Bad | U | 0.029 | 0.017 | 5.63[+] | 0.018 | 0.033 | 0.023 | 0.78[+] | 0.377 |
| | M | 0.029 | 0.028 | 0.14 | 0.888 | 0.033 | 0.025 | 0.44 | 0.664 |
| Very bad | U | 0.003 | 0.003 | 0.20[+] | 0.655 | 0.005 | 0.003 | 0.35[+] | 0.555 |
| | M | 0.003 | 0.003 | 0.14 | 0.890 | 0.005 | 0.008 | −0.26 | 0.796 |
| K6 score | U | 3.120 | 2.850 | 1.77 | 0.076 | 2.957 | 2.923 | 0.12 | 0.904 |
| | M | 3.120 | 3.145 | −0.11 | 0.912 | 2.957 | 2.817 | 0.37 | 0.711 |
| # of children in household | U | 0.643 | 0.786 | −3.99 | 0.000 | 0.788 | 0.808 | −0.30 | 0.762 |
| | M | 0.643 | 0.656 | −0.29 | 0.771 | 0.788 | 0.789 | −0.01 | 0.993 |
| Household size | U | 2.976 | 3.076 | −1.75 | 0.080 | 3.130 | 3.176 | −0.42 | 0.675 |
| | M | 2.976 | 2.991 | −0.19 | 0.849 | 3.130 | 3.151 | −0.14 | 0.890 |
| Ability of daily activity[(i)] | U | 0.083 | 0.062 | 4.29[+] | 0.038 | 0.109 | 0.065 | 5.61[+] | 0.018 |
| | M | 0.083 | 0.084 | −0.04 | 0.966 | 0.109 | 0.102 | 0.20 | 0.839 |
| Exercise[(ii)] | | | | | | | | | |
| Mild | U | 1.565 | 1.488 | 0.97 | 0.330 | 1.152 | 1.239 | −0.64 | 0.525 |
| | M | 1.565 | 1.559 | 0.06 | 0.955 | 1.152 | 1.195 | −0.23 | 0.818 |
| Moderate | U | 1.133 | 1.078 | 0.83 | 0.408 | 0.788 | 0.850 | −0.56 | 0.579 |
| | M | 1.133 | 1.143 | −0.10 | 0.919 | 0.788 | 0.732 | 0.38 | 0.705 |
| Hard | U | 0.220 | 0.210 | 0.33 | 0.744 | 0.120 | 0.119 | 0.01 | 0.990 |
| | M | 0.220 | 0.218 | 0.04 | 0.971 | 0.120 | 0.106 | 0.25 | 0.805 |
| Alcohol[(iii)] | U | 1.071 | 0.978 | 2.31 | 0.021 | 1.174 | 1.020 | 2.08 | 0.038 |
| | M | 1.071 | 1.069 | 0.03 | 0.973 | 1.174 | 1.192 | −0.16 | 0.870 |

*(Continued)*

**Table 3.** (Continued)

| | | Cognitive | | | | Manual | | | |
|---|---|---|---|---|---|---|---|---|---|
| | | Mean | | t-test/χ²test | | Mean | | t-test/χ²test | |
| | | Diagnosed | Not diag. | statistics | p-value | Diagnosed | Not diag. | statistics | p-value |
| Smoking[(iv)] | U | 0.594 | 0.535 | 1.37 | 0.171 | 0.804 | 0.737 | 0.78 | 0.436 |
| | M | 0.594 | 0.606 | −0.18 | 0.858 | 0.804 | 0.793 | 0.10 | 0.924 |
| Log (income) | U | 3.146 | 3.131 | 0.40 | 0.690 | 2.841 | 2.936 | −1.67 | 0.094 |
| | M | 3.146 | 3.164 | −0.32 | 0.745 | 2.841 | 2.833 | 0.10 | 0.922 |
| Log (Household income) | U | 3.865 | 3.859 | 0.23 | 0.818 | 3.659 | 3.633 | 0.58 | 0.559 |
| | M | 3.865 | 3.878 | −0.35 | 0.727 | 3.659 | 3.675 | −0.27 | 0.784 |
| Job category | | | | | | | | | |
| Professional | U | 0.300 | 0.307 | 0.14[+] | 0.705 | | | | |
| | M | 0.300 | 0.301 | −0.03 | 0.976 | | | | |
| Managerial | U | 0.194 | 0.174 | 1.59[+] | 0.208 | | | | |
| | M | 0.194 | 0.194 | 0.02 | 0.980 | | | | |
| Clerical | U | 0.184 | 0.200 | 0.92[+] | 0.338 | | | | |
| | M | 0.184 | 0.185 | −0.04 | 0.972 | | | | |
| Sales | U | 0.127 | 0.115 | 0.78[+] | 0.378 | | | | |
| | M | 0.127 | 0.127 | −0.03 | 0.977 | | | | |
| Service | U | 0.196 | 0.204 | 0.25[+] | 0.618 | | | | |
| | M | 0.196 | 0.194 | 0.07 | 0.945 | | | | |
| Security | U | | | | | 0.098 | 0.097 | 0.00[+] | 0.967 |
| | M | | | | | 0.098 | 0.103 | −0.16 | 0.872 |
| Primary industries [(v)] | U | | | | | 0.076 | 0.052 | 2.22[+] | 0.136 |
| | M | | | | | 0.076 | 0.078 | −0.08 | 0.938 |
| Transport | U | | | | | 0.245 | 0.209 | 1.38[+] | 0.241 |
| | M | | | | | 0.245 | 0.249 | −0.10 | 0.917 |
| Manufacturing | U | | | | | 0.582 | 0.642 | 2.93[+] | 0.087 |
| | M | | | | | 0.582 | 0.570 | 0.23 | 0.817 |
| Risk factor | | | | | | | | | |
| Hypertension | U | 0.388 | 0.306 | 18.36[+] | 0.000 | 0.404 | 0.324 | 5.16[+] | 0.021 |
| | M | 0.388 | 0.391 | -0.09 | 0.930 | 0.404 | 0.406 | -0.04 | 0.972 |
| Dyslipidaemia | U | 0.315 | 0.254 | 11.48[+] | 0.001 | 0.306 | 0.208 | 11.65[+] | 0.001 |
| | M | 0.315 | 0.314 | 0.07 | 0.946 | 0.306 | 0.315 | -0.20 | 0.845 |

In the table, U denotes unmatched, and M denotes matched.

[+]: Chi-square statistics are reported. (Degree of freedom is 1)

[(i)] The ability of daily living was measured using a dichotomous variable that took the value of '1' if the respondent answered 'yes' to the question: 'do you have any problem in your daily life?'

[(ii)] Frequency of exercise is indexed into six categories; 0 = not at all, 1 = once a month, 2 = once a week, 3 = two or three times a week, 4 = four or five times a week, and 5 = almost every day.

[(iii)] The average amount of alcohol consumed (calculated in terms of Japanese sake) when drinking was determined using the following categories: 1 = less than one cup/glass (180 ml), 2 = one to three glass(es), 3 = three to five glasses, 4 = five glasses or more. 0 was allocated to those who did not usually drink (or could not).

[(iv)] Average number of cigarettes smoked per day was indexed as follows; 0 = none at all, 1 = 10 or less, 2 = 11 to 20, 3 = 21 to 30, 4 = more than 30.

[(v)] Primary industries include agriculture, forestry, and fishery.

Overall, compared to the control group, the treatment group tended to be older, single, have lower than university-level education, worse SRH, more distressed, a lower number of children and smaller household size, to be more disabled, and have a greater risk of being diagnosed with hypertension and dyslipidaemia. Since confounding factors could cause bias in our

estimates, we adjusted the mean difference between the treatment and control groups using PSM. After applying PSM, we confirmed that the individual characteristics were balanced between the two groups. Note that Table A in the web appendix shows the covariate balance for the analysis of the simultaneous setting; however, the basic statistics for the one-year lagged setting were almost identical (not shown). Other results concerning covariate balancing are also reported in detail in the web appendix.

## Results of PSM and logistic regression

Table 4 shows the ATT and its 95% confidence intervals (CIs), and the odds ratio estimated by logistic regression is reported in Table 5. Further, Fig 3 depicts the ATT with its 95% CIs for Model 1 (male versus female) and Model 2 (cognitive versus manual) for the one-year lagged and simultaneous settings, respectively. In the same way, Fig 4 depicts the odds ratio and its 95% confidence interval. The ATT and odds ratio obtained through the (i)–(iv) procedure is denoted as the 'one-year lagged' effect, and also, the ATT and odds ratio obtained through the (v)–(viii) procedure is denoted as the 'simultaneous' effect.

Overall, the result of Tables 4 and 5 suggests that the effect of cancer on job-quitting likelihood is highly acute, but that the statistical significances for one-year lagged effects vary across gender and type of job.

**Gender gap [Model 1 in Tables 4 and 5].** Firstly, we observe from the first column of Table 4 that the male cancer patients were 5.0 percentage points (95% CI [1.5, 8.5]) more likely to quit their job in the next year of being diagnosed with cancer, and 10.1 percentage points (95% CI [6.9, 13.4]) more likely to quit during the year of diagnosis. On the other hand, the result in the second column suggests that the female cancer patients were 18.6 percentage points (95% CI [13.1, 24.0]) more likely to quit their job during the year they were diagnosed with cancer; however, the effect became statistically insignificant and totally disappeared for the following year (−0.4 percentage points; 95% CI [-5.1, 4.4]). Overall, the results of Model 1

**Table 4. Effects of a cancer diagnosis on the risk of work cessation in terms of gender and type of job (the result of PSM).**

| | Model 1 | | | | | | Model 2 | | | | | |
| | Male | | | Female | | | Cognitive | | | Manual | | |
| | ATT | 95% CI | p-value | ATT | 95% CI | p-value | ATT | 95% CI | p-value | ATT | 95% CI | p-value |
|---|---|---|---|---|---|---|---|---|---|---|---|---|
| Panel (A) | | | | | | | | | | | | |
| one year lagged | 0.050*** | [0.015, 0.085] | 0.005 | −0.004 | [−0.051, 0.044] | 0.864 | 0.038** | [0.002, 0.074] | 0.037 | 0.021 | [−0.040, 0.081] | 0.503 |
| | (0.018) | | | (0.024) | | | (0.018) | | | (0.031) | | |
| Total Observations | 44,241 | | | 32,640 | | | 53,348 | | | 16,385 | | |
| Panel (B) | | | | | | | | | | | | |
| Simultaneous | 0.101*** | [0.069, 0.134] | <0.001 | 0.186*** | [0.131, 0.240] | < 0.001 | 0.116*** | [0.085, 0.147] | <0.001 | 0.187*** | [0.121, 0.253] | < 0.001 |
| | (0.017) | | | (0.028) | | | (0.016) | | | (0.034) | | |
| Total Observations | 53,535 | | | 40,617 | | | 64,682 | | | 20,995 | | |

Bootstrapping standard errors with 200 replications are reported in parentheses.

Inference:

*** p < 0.01;

** p < 0.05;

* p < 0.1.

**Table 5. The odds ratio of a cancer diagnosis on the risk of work cessation in terms of gender and type of job (the result of logistic regression).**

| | Model 1 | | | | | | Model 2 | | | | | |
|---|---|---|---|---|---|---|---|---|---|---|---|---|
| | Male | | | Female | | | Cognitive | | | Manual | | |
| | OR | 95% CI | p-value | OR | 95% CI | p-value | OR | 95% CI | p-value | OR | 95% CI | p-value |
| Panel (A) | | | | | | | | | | | | |
| one year lagged | 1.765*** | [1.283, 2.426] | <0.001 | 1.045 | [0.629, 1.735] | 0.866 | 1.624*** | [1.183, 2.230] | 0.003 | 1.246 | [0.662, 2.344] | 0.495 |
| | (0.287) | | | (0.270) | | | (0.263) | | | (0.402) | | |
| Total Observations | 44,241 | | | 32,692 | | | 53,348 | | | 17,065 | | |
| Log-likelihood | -10060.51 | | | -9269.51 | | | -13010.72 | | | -4422.90 | | |
| Panel (B) | | | | | | | | | | | | |
| Simultaneous | 2.581*** | [2.032, 3.277] | <0.001 | 4.072*** | [3.171, 5.230] | <0.001 | 2.997*** | [2.424, 3.707] | <0.001 | 4.016*** | [2.814, 5.731] | <0.001 |
| | (0.314) | | | (0.520) | | | (0.325) | | | (0.729) | | |
| Total Observations | 53,535 | | | 40,617 | | | 64,682 | | | 20,995 | | |
| Log-likelihood | -11803.64 | | | -11736.53 | | | -15633.38 | | | -5380.97 | | |

Robust standard errors for heteroscedasticity are reported in parentheses.

Inference:

*** p < 0.01;

** p < 0.05;

* p < 0.1.

imply the significant difference in job cessation patterns between men and women. The upper panel of Fig 3 depicts the ATT and its 95% confidence interval for Model 1 so as to be consistent with Table 4.

**Job-type gap [Model 2 in Tables 4 and 5].** For cognitive workers, the result in the third column suggests the average lagged effect remained statistically significant (3.8 percentage points; 95% CI [0.2%, 7.4%]), and the average simultaneous treatment effect was 11.6 percentage points (95% CI [8.5%, 14.7%]). On the other hand, we observe from the final column of Table 3 that manual workers were 18.7 percentage points (95% CI [12.1%, 25.3%]) more likely to quit their current job during the year they were diagnosed with cancer; however, the effect became statistically insignificant and completely vanished in the following year (2.1 percentage points; 95% CI [−0.4%, 8.1%]), which seems to be quite similar to the effect of cancer on female workers. Again, the result of Model 2 implies the huge difference between cognitive workers and manual workers. The lower panel of Fig 3 depicts the ATT and its 95% confidence interval for Model 2 so as to be consistent with Table 3.

**Logistic regression using the full sample.** In addition to the sub-sample analyses implemented in the above part, it is possible and interesting to estimate the difference in work cessation risk between men and women or manual and cognitive workers (regardless of cancer diagnosis). To evaluate this effect, we consider running the following logistic regression.

$$p(quit) = \frac{\exp(\beta_0 + \beta_1 Diag_{it} + \beta_2 Women_{it} + \beta_3 Manual_{it} + \gamma X_{it} + \delta \theta_t + \epsilon_{it})}{1 + \exp(\beta_0 + \beta_1 Diag_{it} + \beta_2 Women_{it} + \beta_3 Manual_{it} + \gamma X_{it} + \delta \theta_t + \epsilon_{it})}, \quad (4)$$

where the coefficient $\beta_2$ ($\beta_3$) measures the relative risk between men and women (manual and cognitive worker). Table 6 reports the odds ratio of each factor ($\exp(\beta_1)$, $\exp(\beta_2)$ and $\exp(\beta_3)$) and its 95% confidence interval.

From the above table, we can see that the cancer diagnosis has a huge effect on the decision of quitting the job. Moreover, the result suggests that female workers and manual workers are likely to suffer from a higher risk of job cessation regardless of the cancer diagnosis.

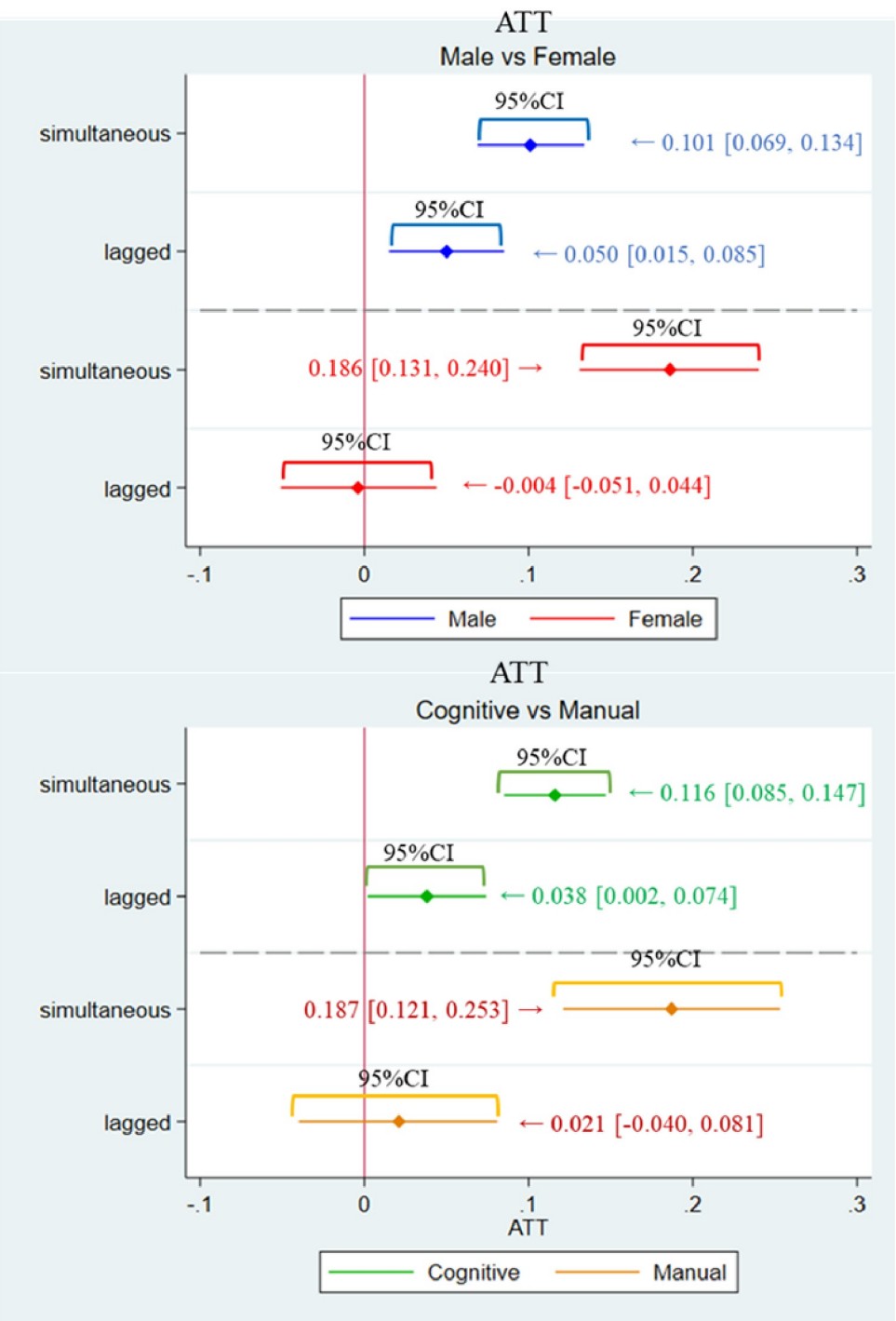

**Fig 3. ATT (estimates and 95% confidence intervals, which corresponds to the result in Table 4).**

Specifically, the probability that female (manual) workers quit their job at a specific period $t$ given that she worked in period $t − 1$ would 1.234 (1.074) times higher than that of male (cognitive) workers, and the probability that female workers quit their job at a specific period $t + 1$ given that she worked in period $t − 1$ and $t$ would be 1.279 (1.075) times higher than that of male (cognitive) workers.

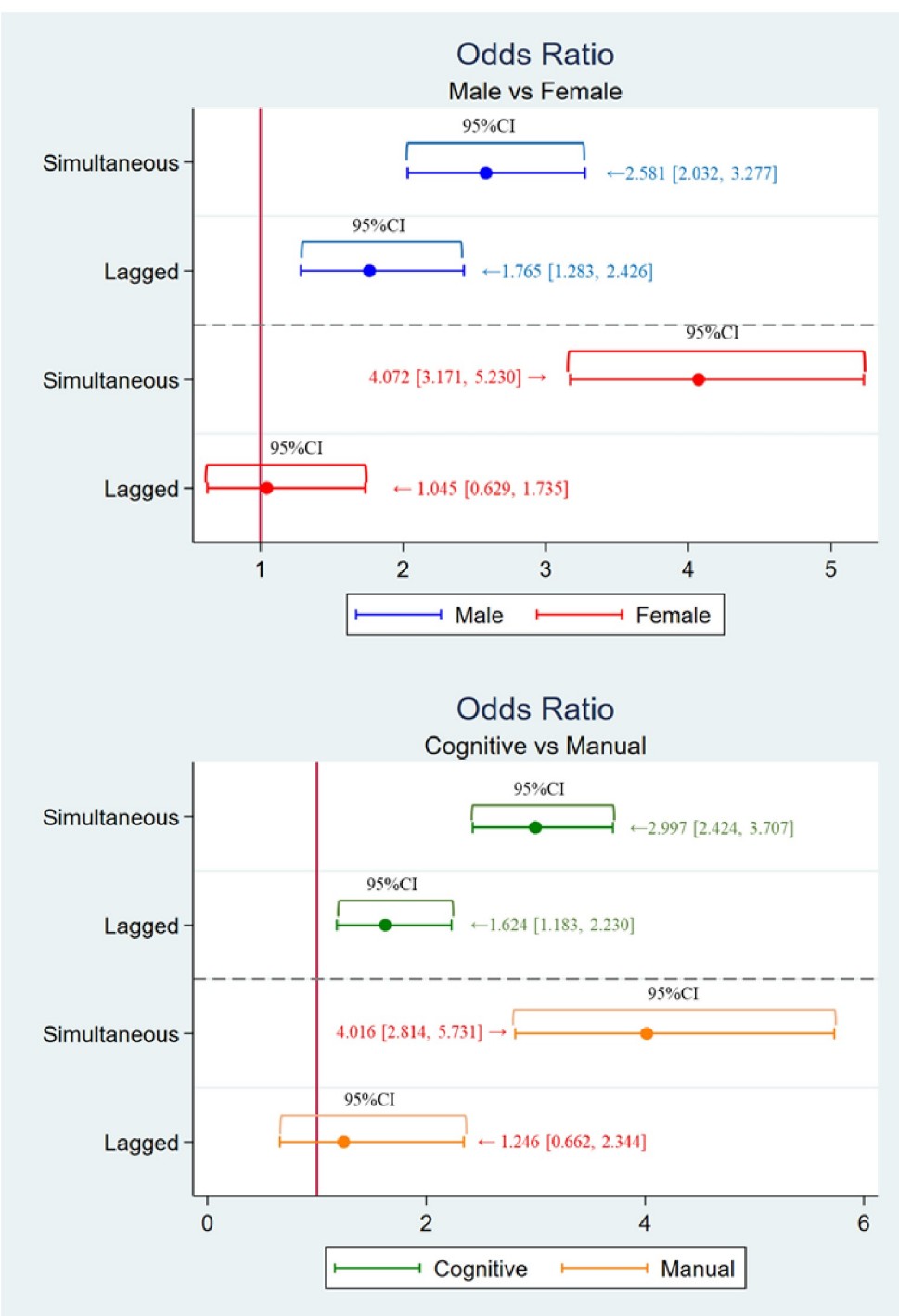

**Fig 4. Odds ratio (estimates and 95% confidence intervals, which corresponds to the result in Table 5).**

## Conclusion

### Discussion

This study sought to determine whether the risk of work cessation after a cancer diagnosis is impacted by gender and job type. To this end, we scrupulously constructed an estimation

Table 6. The result of logistic regression (full sample is used).

| | (1) | | (2) | |
| | Simultaneous | | One year lagged | |
| | OR | 95% CI | OR | 95% CI |
|---|---|---|---|---|
| $Diag_{it}$ | 3.047*** | [2.570, 3.614] | 1.551*** | [1.189, 2.022] |
| | (0.265) | | (0.210) | |
| $Women_{it}$ | 1.224*** | [1.140, 1.315] | 1.269*** | [1.171, 1.374] |
| | (0.045) | | (0.052) | |
| $Manual_{it}$ | 1.070** | [1.008, 1.135] | 1.076** | [1.008, 1.149] |
| | (0.032) | | (0.036) | |
| Observation | 107,899 | | 88,955 | |
| Log Likelihood | -24542.53 | | -20254.85 | |

Robust standard errors for heteroscedasticity are reported in parentheses. All other covariates used as matching information are controlled.

Inference:

*** $p < 0.01$;

** $p < 0.05$;

* $p < 0.1$.

framework to identify causal relationships. Although a myriad of previous studies have revealed the strong negative effect of cancer diagnosis and/or cancer treatment on worker's life status [2, 4–7], our study succeeded to extend this line of research by estimating "causal" effect, which was not fully captured in prior literatures, along with PSM. Our findings can be summarised as follows. Firstly, the simultaneous effect of the diagnosis of cancer on work cessation seems to be statistically robust, regardless of gender and type of job, which is highly consistent with the findings of myriad previous studies [25,26]. Secondly, we found a possibility of a gender difference regarding labour continuance after a cancer diagnosis. Similarly, individuals who engaged in manual work showed a more serious risk of work cessation upon diagnosis of cancer.

Compared to the survey conducted in 2004 where 34% of newly diagnosed cancer patients had to quit their job or were dismissed [10], our study suggests that the working environment for cancer patients has become improved. However, even if this is the case, our result underlines the fact that the gender/ job type gap has not been solved.

What causes gender differences in regard to the effect of cancer on work cessation? An important, implication is that females are more likely to be marginalized in the workplace than males. Although there have been some acts or guidelines [27] to abolish the discrimination based gender in Japan, our results, unfortunately, suggest that female workers are more prone to face the implicit pressure to cease their job after some health shock. The "observable" statistics (e.g. wage differences) appear to be improved [28]; however, our results corroborate that the change in the "unobservable" characteristics (e.g. prejudice, value) takes longer time.

Regarding differences in terms of types of jobs, although the tendency seems to be similar, the source of the differences should not be the same (because only 20.7% of women are working as manual workers, in contrast to 44.6% of men [29]). One major and straightforward reason for this discrepancy is that occupations classified as manual work often involve notable physical burdens. Consequently, physical constraints caused by disease or medical treatment mean that manual workers can be less able to manage such physical burden and, thus, be more likely to quit their jobs than are cognitive workers. This fact might imply that the system for

supporting patients is still immature, especially for those engaging in physically demanding tasks.

While the MHLW has developed guidelines for the realisation of a society in which individuals can continue working while receiving treatment for cancer, our analyses provide some evidence that this policy is still its infancy as regards gender and job-type differences. As a result, our study can contribute to the research field in that we visualise and compare the negative effect of cancer on work cessation and show that there are still some notable obstacles at the workplace for female and manual workers. Furthermore, our results provide some benchmarks for future studies, because we provide a reliable estimate of 'causal' effects, considering endogeneity issues caused by confounding factors.

## Limitations

Our statistical strategy is limited in some respects. Firstly, PSM cannot address unobserved and time-variant individual heterogeneous characteristics. For instance, our results lose their reliability if the preference to work is changed, which significantly affects decisions to quit working over time. However, thanks to the opulence of the number of variables in LSMEP data, the influence of unobserved and time-variant factors should be mitigated, to some extent. Secondly, the method we employ in PSM cannot estimate the precise causal effect. As we noted, because our two strategies trade-off with each other, it is impossible to identify the true effect. On the other hand, however, the most plausible situation in which people are diagnosed with cancer after work cessation (reversal direction) is after they reach their mandatory retirement age in their workplace. As we performed the model with a dichotomous variable, taking the value of '1' if age is 60 years or older, which is the typical mandatory retirement age in Japan, and we obtained similar results (which are available on request), it is unlikely that our analysis suffers from a reverse-causality issue. Finally, due to the data limitation, we were not able to identify types of cancer. Because a prognosis would be divergent depending on types [30], our result should be interpreted with caution in that it does not control for these types. Of course, this limitation hinders us from deriving the implication from the clinical viewpoint; however, our study captures the average effect of cancer diagnosis on the labor market by focusing on the risk of job cessation.

## Concluding remarks

In this research, we determined that the risk of work cessation after a cancer diagnosis is impacted by gender and job type. In the analyses conducted by PSM, we observe that the male (female) cancer patients were 5.0 (-0.4 but insignificant) percentage points more likely to quit their job in the next year of being diagnosed with cancer, and 10.1 (18.6) percentage points more likely to quit during the year of diagnosis. Regarding the job type gap, the one year lagged effect on cognitive (manual) workers are 3.8 (2.1 but insignificant) percentage points and the simultaneous effects amount to 11.6 (18.7) percentage points.

Furthermore, we showed that opportunity cost can be a strong trigger for work cessation among cancer patients, and that a system that helps them to continue working while receiving medical treatment can play a crucial role in keeping such individuals in the labour market.

The present study also presents a milestone for future research. Our analysis did not take the welfare of workers into account; therefore, it is unclear whether continuing working after diagnosis is truly better for workers' overall utility. For instance, in a society that does not have a national insurance system (e.g., the United States), patients may continue working in order to avoid losing health insurance, regardless of their own wishes. In Japan, this is not the case; however, as the cost of medical treatment for non-communicable diseases (NCDs) can be a

significant burden, it is likely that such individuals will continue to work. Further studies that investigate this relationship from an economic and welfare perspective should be warranted.

## Supporting information

**S1 Appendix.**
(PDF)

## Author Contributions

**Conceptualization:** Shuhei Kaneko, Haruko Noguchi, Rong Fu, Cheolmin Kang, Shinsuke Amano, Atsushi Miyawaki.

**Formal analysis:** Shuhei Kaneko.

**Funding acquisition:** Haruko Noguchi.

**Methodology:** Shuhei Kaneko.

**Project administration:** Haruko Noguchi.

**Supervision:** Haruko Noguchi, Atsushi Miyawaki.

**Writing – original draft:** Shuhei Kaneko.

**Writing – review & editing:** Haruko Noguchi, Rong Fu, Cheolmin Kang, Akira Kawamura, Atsushi Miyawaki.

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
