## [Decision Letter · Decision Letter 0]

4 Nov 2019

PONE-D-19-23528

Differences in Cancer Patients’ Work-Cessation Risk, based on Gender and Type of Job: Examination of Middle-Aged and Older Adults in Super-Aged Japan

PLOS ONE

Dear Dr. Kaneko, 

Thank you for submitting your manuscript to PLOS ONE. After careful consideration, we feel that it has merit but does not fully meet PLOS ONE’s publication criteria as it currently stands. Therefore, we invite you to submit a revised version of the manuscript that addresses the points raised during the review process.

We would appreciate receiving your revised manuscript by Dec 19 2019 11:59PM. To enhance the reproducibility of your results, we recommend that if applicable you deposit your laboratory protocols in protocols.io, where a protocol can be assigned its own identifier (DOI) such that it can be cited independently in the future. For instructions see: http://journals.plos.org/plosone/s/submission-guidelines#loc-laboratory-protocols

We look forward to receiving your revised manuscript.

Kind regards,

Jason Chia-Hsun Hsieh, M.D. Ph.D

Academic Editor

PLOS ONE

Journal Requirements:

2. In ethics statement in the manuscript and in the online submission form, please provide additional information about the patient records used in your retrospective study. Specifically, please ensure that you have discussed whether all data were fully anonymized before you accessed them and/or whether the IRB or ethics committee waived the requirement for informed consent. If patients provided informed written consent to have data from their medical records used in research, please include this information.

This study was partly funded by generous support from Waseda University Research Initiatives, entitled ‘Empirical and Theoretical Research for Social Welfare in Sustainables Society - Inheritance of human capital beyond “an individual” and “a generation”’, and in part by a grant-in-aid for a scientific research project from the Ministry of Health, Labour and Welfare (H29- Junkankitou-Ippan-002), entitled ‘Effects of the Prevention Policy of Lifestyle-related Disease on Labor Productivity and Macro Economy from Viewpoint of Cost-effective Analysis’

HN is funded by funded by the Japanese Ministry of Health, Labour and Welfare (H29-Junkankitou-Ippan-002), and Organization for University Research Initiatives of Waseda University.

RF is funded by Japan Society for the Promotion of Science (JSPS KAKENHI Grant Number 17H07182).

The funders had no role in study design,

data collection and analysis, decision to publish, or preparation of the manuscript.

Additional Editor Comments:

The article and research topic is interesting; however, authors require to answer several essential questions.

Reviewers' comments:

Reviewer's Responses to Questions

**Comments to the Author**

1. Is the manuscript technically sound, and do the data support the conclusions?

Reviewer #1: Yes

Reviewer #2: Yes

Reviewer #3: No

Reviewer #5: Partly

2. Has the statistical analysis been performed appropriately and rigorously? 

Reviewer #1: Yes

Reviewer #2: Yes

Reviewer #3: No

Reviewer #5: No

3. Have the authors made all data underlying the findings in their manuscript fully available?

Reviewer #1: No

Reviewer #2: Yes

Reviewer #3: Yes

Reviewer #5: Yes

4. Is the manuscript presented in an intelligible fashion and written in standard English?

Reviewer #1: Yes

Reviewer #2: Yes

Reviewer #3: Yes

Reviewer #5: Yes

5. Review Comments to the Author

Reviewer #1: This study tried to estimate the effect cancer diagnosis has on labour-force

Participation. Author answered this question by propensity score rather than other common statistic methods, such as logistic regression or survival analysis. It is designed to avoid multiple confounding factor and possible reverse causal effect. The concept is OK and the matching method is clear. However, I have some suggestions.

1. What is the difference between table 3 and figure 3? They contained almost the same information. The valuable data is estimation of ATT and 95% confidence of ATT. Author may consider the redundancy in data presentation.

2. The major endpoint in this study is average treatment effect on the treatment group (ATT) which was derived from propensity score. However, I am a little confused. To my best of knowledge, propensity score is the value of log odds given particular conditional variables. In table 3, the simultaneous ATT in female is 0.186. Does that really mean the female cancer patients were 18.6% more likely to quit their job? Or the relative risk is e^0.186=1.204. Then exact female cancer patients were 20.4% more likely to quit their job. I am not sure if I misunderstood the author’s reporting. Since ATT is the main concept in this study, author should add more detail process of data estimation as well as clinical implications. In addition, how big the ATT is denote clinical significant impact?

3. There is no mention about statistical method and software. Author should add more detailing process.

Reviewer #2: The authors evaluated the effect of cancer diagnosis on labour-force participation among middle-aged and older population in Japan. Female workers are more likely to quit their job immediately than male. Manual workers are more prone to quit their job. Their results are partially interesting, however the analysis is insufficient to reach their conclusion and discussion. Our concerns are as follows.

1. In Table 2, there are some statistical difference in age, marital status, education level and so on. Although they adjusted these confounding factors in their analysis model, these factors might be important to decide to quit the job. Thus, further analysis will allow us the novel discussion in this study.

2. Although they mentioned in study limitation, these results might be affect by severity of diseases, choice of treatment (surgery or medicine) or several medical aspects. In terms of clinical viewpoint, this analysis is insufficient to reach the conclusion.

3. It is interesting to find the gender gap and job-type gap in the analysis. Could you add the further analysis, such as male cognitive worker vs. male manual workers and female cognitive worker vs. female manual workers.

Reviewer #3: Manuscript number: PONE-D-19-23528

Title: Differences in Cancer Patients’ Work-Cessation Risk, based on Gender and Type of Job: Examination of Middle-Aged and Older Adults in Super-Aged Japan

Reviewer comments:

This paper was a nationwide population-based longitudinal survey estimate the effect cancer diagnosis has on labour-force participation among middle-aged and older populations in Japan. The following suggestions are intended to help the authors revise this paper for future publication and to assist them in preparing future work.

1. Please add the cancer epidemiology in Japan.

2. Please strength the significance and importance of this issue in introduction section.

3. Are there preliminary data or previous relate studies regrading the work-cessation in Japan, if possible, please add in the introduction section.

4. Please add the research purpose at the end of introduction section.

5. Please revise the discussion section following the revising introduction and the findings of the current study compare or discuss with previous studies, in which cited in the introduction section.

6. Conclusion was general, please revise following the findings of this current study.

7. Abstract: please revise the results following the research purpose and revise the conclusion following the revising conclusion in the texts.

Reviewer #5: The authors conducted a population based study using the annual survey data to estimate the effect cancer diagnosis has on labour-force participation among middle-aged and older populations in Japan. Overall, they identified that the diagnosis of cancer has a huge effect on labour-force participation among the population, but this effect varies across subpopulations. I have some major concerns on the statistical evaluation.

1. There are several major errors for the statistical testing and evaluation. First, in Table 2, the comparisons between diagnosed vs. not diagnosed on different demographic variables were conducted using t-test. It is relevant for continuous variables such as age and income, but not correct for categorical variables such as marital status, SRH, etc. The suitable statistical test should be chisq test to compare categorical variable between two subgroups.

2. Second, to evaluate the work cessation between male and female, manual and cognitive, logistic regression model can be considered while gender, and type of job can be treated as predictors. The advantage is that, it will provide odd ratio estimation to evaluate the magnitude of the differences. Besides that, it can also incorporate other variables to conduct multivariable analysis which was not conducted in the study.

3. There are additional analyses can be conducted to explore the potential interactive effect between gender and other factors, and type of job and other factors, on work cessation.

4. It would be interesting to explore the working cessation rate after diagnosis across different year period.

5. It would be interesting to know the major types of cancer that were diagnosed and their impact on work cessation.

6. PLOS authors have the option to publish the peer review history of their article (what does this mean?). If published, this will include your full peer review and any attached files.

Reviewer #1: No

Reviewer #2: No

Reviewer #3: No

Reviewer #5: No

---

## [Author Response · Author response to Decision Letter 0]

23 Dec 2019

Please refer to the response letters.

---

## [Editor Report · Decision Letter 1]

31 Dec 2019

Differences in Cancer Patients’ Work-Cessation Risk, based on Gender and Type of Job: Examination of Middle-Aged and Older Adults in Super-Aged Japan

PONE-D-19-23528R1

Dear Dr. Kaneko, 

We are pleased to inform you that your manuscript has been judged scientifically suitable for publication and will be formally accepted for publication once it complies with all outstanding technical requirements.

With kind regards,

Jason Chia-Hsun Hsieh, M.D. Ph.D

Academic Editor

PLOS ONE

Additional Editor Comments (optional):

All the questions were answered adequately.
---

## [Editor Report · Acceptance letter]

3 Jan 2020

PONE-D-19-23528R1 

Differences in Cancer Patients’ Work-Cessation Risk, based on Gender and Type of Job: Examination of Middle-Aged and Older Adults in Super-Aged Japan 

Dear Dr. Kaneko:

I am pleased to inform you that your manuscript has been deemed suitable for publication in PLOS ONE. Congratulations! Your manuscript is now with our production department. 

With kind regards,

on behalf of

Dr. Jason Chia-Hsun Hsieh 

Academic Editor

PLOS ONE